# Roles of Neutrophils in Autoimmune Diseases and Cancers

**DOI:** 10.3390/ijms26189040

**Published:** 2025-09-17

**Authors:** Anjali Bhargav, Vinay Kumar, Neeraj Kumar Rai

**Affiliations:** 1Department of Clinical Immunology and Rheumatology, Sanjay Gandhi Post Graduate Institute of Medical Sciences, Lucknow 226014, India; anjbhargav@gmail.com; 2Hershey Medical Center, Heart and Vascular Institute, Pennsylvania State University, Hershey, PA 17033, USA; vinayktyagi07@gmail.com; 3Nora Eccles Harrison Cardiovascular Research and Training Institute, University of Utah, Salt Lake City, UT 84112, USA

**Keywords:** neutrophil, neutrophil extracellular traps, autoimmune diseases, cancer, systemic lupus erythematosus, rheumatoid arthritis, juvenile idiopathic arthritis, low density neutrophils, tumor-associated neutrophils, anti-tumorigenic neutrophils, pro-tumorigenic neutrophils

## Abstract

Neutrophils, a first-line defender, has a multifaceted presence in chronic inflammation, autoimmune pathology, and tumor progression. The microenvironmental cues facilitate functional plasticity and phenotypic heterogeneity to neutrophils that enable both their protective and pathogenic roles. Autoimmune diseases including systemic lupus erythematosus (SLE), rheumatoid arthritis (RA), and juvenile idiopathic arthritis (JIA) display the presence of dysregulated subsets of neutrophil, such as low-density granulocytes (LDGs) that promote proinflammation and contribute to tissue damage via NETosis and type I interferon-mediated signaling. In cancer, particularly tumors, they exhibit tumor-associated neutrophils (TANs) which may polarize either towards anti-tumorigenic ‘N1’ or pro-tumorigenic ‘N2’ phenotypes based on available modulators such as TGF-β and leucine-driven epigenetic modifications. The development in neutrophil biology has introduced several novel therapeutic strategies that allow NET targeting, inhibition of chemokine receptors like CXCR2, and exploration of neutrophil-derived biomarkers for diagnosis and disease monitoring. Such findings encourage the importance of neutrophils as both effectors and therapeutic targets in inflammatory and neoplastic conditions.

## 1. Introduction

Neutrophils are granulocytes, the most abundant myeloid cell type in the vascular system. Their differentiation and maturation initiate in the bone marrow before entering the blood circulation. This process is regulated through an intricate pathway via several stages of hematopoietic stem cells (HSCs) that exhibit signature gene expression. During development, three neutrophil subsets reside in bone marrow; a proliferative precursor of neutrophils named pre-neutrophils (preNeus), an intermediate neutrophil (immature Neus), and mature neutrophils population [1]. The preNeus are early-stage neutrophils that do not respond to infection and inflammation. At the same time, immature Neus enter the bloodstream late and respond to the immune response less efficiently than mature Neus. Mature neutrophils lose self-renewal ability before entering blood circulation and exhibit an immune response against infection. They also play a decisive role in excluding pathogens in hosts through their antimicrobial functions, such as phagocytosis, degranulation, NET (neutrophil extracellular traps) development, and ROS (reactive oxygen species) generation (Figure 1).

Apart from their function in innate immunity, neutrophils also regulate the adaptive immunity effectively by migrating to lymphoid organs, presenting antigens, and modulating the lymphocyte responses [2,3]. Neutrophils’ profound role in innate and adaptive immunity exerts its dynamic and plastic properties with distinct subsets. However, neutrophils do not contribute exclusively to providing immunity against antigens. It may shift toward promoting chronic inflammation and immune dysregulation through several mechanisms, like NET formation (NETosis) and dysfunctional neutrophil subsets. NETosis has an identified role in host defense, but dysregulated NETosis exposes modified self-antigens, stimulates type I interferon production, and plasmacytoid dendritic cells, with activation that further activates APC (antigen-presenting cells) and T cells mediated by an inflammatory loop (Figure 2) [4]. This phenomenon drives the pathogenesis of autoimmune conditions like systemic lupus erythematosus (SLE) [5], rheumatoid arthritis (RA) [6], and type 1 diabetes (T1D) [7]. Moreover, patients with autoimmune disease often exhibit elevated low-density neutrophils (LDNs) and low-density granulocytes (LDGs) subsets with increased antigen-presenting capacity, NETosis, and proinflammatory activity (Figure 2) [7]. Along with these effector activities, hyperactive or dysregulated neutrophil subsets may exacerbate autoimmune responses and tissue damage. Besides NETs, calprotectin and neutrophil serine proteinases are other neutrophil-based biomarkers that are correlated to autoimmunity. Neutrophils are also able to recruit and activate a variety of immune and nonimmune cells that contribute to regulating inflammation [8].

Interestingly, while neutrophils exhibit antigen clearance and protect host immunity, altered signaling within the tumor microenvironment (TME) reprograms them to adopt such a function and phenotype that encourages tumor-promoting activities. Dual and often contradictory roles of neutrophils are reported; for example, ROS [9] or tumor necrosis factor-related apoptosis-inducing ligand (TRAIL)-mediated apoptosis can stimulate T cell responses [10] that exhibit an anticancer effect (Figure 2). Conversely, tumor-associated neutrophils (TANs) [11] and pro-tumor phenotype [12] of neutrophils support tumor progression by promoting angiogenesis, suppressing cytotoxic T cells, and facilitating metastasis via extracellular matrix remodeling and epithelial–mesenchymal transition (EMT) (Figure 3).

Recent studies have identified distinct pro-and anti-tumor neutrophil phenotypes that can transform themselves among each other under the influence of different local cues associated with specific signaling pathways and functional markers (Table 1). Influencing factors that enable such pro-tumor to anti-tumor transitions and vice versa are not limited to prognosis through distinct neutrophil biomarkers but also have potential therapeutic targets. Along with neutrophils, NETs themselves secrete various molecules, like self-DNA and other signaling factors that enable the crosstalk with other immune cells, such as T cells, B cells, natural killer (NK) cells, and plasmacytoid dendritic cells (pDCs), which help them to modulate adaptive immune responses based on NET immunogenicity. Therefore, inhibiting NET formation, modulating neutrophil recruitment, or reprogramming neutrophil phenotypes are active areas of therapeutic exploration in both autoimmune and cancer [13,14].

## 2. Neutrophil Biology: Maturation, Subtypes, and Functional Plasticity

The developing neutrophil biology uncovers the extended subsets that exhibit distinct morphology, functionality, and gene expression. This diverse neutrophil population impacts the immune system distinctly based on regulating inflammatory responses and adaptability to cellular microenvironment. For instance, a long time ago, Pillay J et al. identified a subset of human neutrophils with unique phenotypes and functionality, characterized by hyper-segmented nuclear morphology. These neutrophils were comparatively more mature than normal blood neutrophils. They respond to severe inflammation even when T cell activity is suppressed. These mature neutrophils display a distinctive surface marker profile (CD62Ldim/CD16bright/CD11bbright/CD54bright) whereas short-lived classic neutrophils show (CD62Ldim/CD16bright/CD11bbright/CD54dim) [18]. CD54 (ICAM) neutrophils show increased reverse trans-endothelial migration and NETosis under the influence of inflammatory signals, contributing to chronic systemic inflammation. CD54high or CD54bright exhibit a more proinflammatory feature than CD54dim and present their dysregulation [19]. Another surface marker, CD11b, represents active adhesion and migration, expressed by myeloid cells including neutrophil, macrophage, and monocytes. It resides stably within secretory granules of neutrophils and is expressed at a minimal level on the surface along with other neutrophil markers such as CD16 [20]. The elevated ROS and chronic inflammation upregulate CD11b expression on normal neutrophils and highly express on dysregulated subsets of neutrophils (Figure 1) [21].

Beyond the surface marker subsets, neutrophils are classified into functionally distinct populations; for example, classic neutrophils, low-density neutrophils (LDNs), tumor-associated neutrophils (TANs), and G-CSF-primed neutrophils. The classic neutrophils are known for their innate immune response against microbial infections. They eliminate pathogens via phagocytosis of bacterial cells, liberation of antimicrobial peptides, and ROS production. LDNs are a neutrophil subtype observed in the peripheral blood of patients diagnosed with conditions like cancer, autoimmune diseases, sepsis, and other inflammatory disorders. Their size and density are less than those of classic neutrophils. These cells are known for suppressing T cell responses and tumor promotion. Importantly, LDN’s population coexists with immature MDSCs (myeloid-derived suppressor cells; PMN-MDSCs) and mature cells [22]. CD11b and CD16 are common markers of LDNs (Figure 1). CD36, CD41, CD61, and CD226 upregulated on LDNs associated with non-small-cell lung cancer (NSCLC) [23] and CD10, CD16, and CD45high present a mature phenotype with T cell-suppressive role in blood cancer [24]. Elevated NETosis is one of the functional features of LDNs or low-density granulocytes (LDGs) in SLE and severe inflammation. LDGs are also referred to as LDNs, a proinflammatory neutrophil subset that co-exists with peripheral blood mononuclear cells and separate during density gradient centrifugation in various immune-mediated diseases [25].

TANs (tumor-associated neutrophils) are a neutrophil population that accumulate within the tumors and exhibit tumor progression, possibly by compromising immune response and enabling angiogenesis. It contributes to immune evasion by regulatory T cell (Treg) expansion and arginase-1 and other immunosuppressive molecules expression [26]. High levels of CCL2, CXCL2, CCL8, IL-8/CXCL8, and CXCL16 may characterize it. Another subset of TANs exhibits anti-tumor features, characterized by TNFα, CCL3, and ICAM-1. Anti-tumor TANs produce ROS, hydrogen peroxide (H_2_O_2_), Nitric oxide (NO), and neutrophil elastase (NE) that induce oxidative stress and membrane damage. Further, stressed cells go through Fas ligand-receptor (Fas ligand expressed on neutrophil binds to receptor on tumor cell)-based, ADCC (Antibody-dependent cell-mediated cytotoxicity), TRAIL pathway- and CTL (Cytotoxic T lymphocyte)-based apoptosis in tumor cells [5,27]. G-CSF-primed neutrophils have been noticed as a response to G-CSF (Granulocyte colony-stimulating factor; a cytokine) treatment as it can stimulate neutrophil production from bone marrow. It may help in neutrophil recovery by increasing their population while in an emergency, under the regulation of IL-17 [28]. G-CSF-primed neutrophils are featured with over-expression of adhesion molecules, and activation markers of phagocytosis [29]. Overall, the shape of neutrophils, their adaptation to maturation state, surrounding microenvironment, and disease context significantly influence the neutrophil behaviors. This remarkable heterogeneity and functional plasticity underlie their diverse role in both autoimmune diseases and cancer.

## 3. Neutrophils in Autoimmune Diseases

### 3.1. Systemic Lupus Erythematosus (SLE)

Dysregulated neutrophil subsets play a key role in pathogenesis of systemic lupus erythematosus due to their dysfunction such as ROS generation and granule proteases release that results in vascular tissue damage and trigger NET formation simultaneously. NETs expose neoepitopes, to which immune cells bind, these may nuclear autoantigens such as double-stranded DNA and modified histones. These antigen-specific peptides bind to B cell receptors (BCR) and stimulate the autoantibody production, including anti-dsDNA and anti-histone antibodies, followed by B cell activation, regulated by B cell-activating factor (BAFF) and interferons (IFNs) that sustain autoimmunity. In maintaining homeostasis, neutrophils undergo spontaneous apoptosis, but this process can be accelerated. Enhanced neutrophil apoptosis increases the apoptotic burden, generating antinuclear autoantibodies. Dysregulated NETs released by neutrophils also secrete proinflammatory mediators that recruit other immune cells to contribute tissue damage and a prominent subset of neutrophils in chronic inflammation called proinflammatory neutrophils [4,30]. Through the diverse subsets of neutrophils, LDGs expand significantly in SLE and are characterized by a distinct proinflammatory phenotype. Recently, Saisorn et al. reported highly CD66b-expressing LDGs that upregulate adhesion, NET formation, and apoptosis during lupus pathogenesis. In addition, the serum levels of IL-10 and TNF-α are significantly elevated in the patients suffering from childhood-onset SLE. Additionally, the presence of extracellular traps (ETs) and serum citrullinated histone H3 (CitH3) in LDGs compared to healthy controls may serve as practical biomarkers for disease severity. NETs act as potent autoantigens, capable of triggering autoantibody production and driving a robust type I interferon response that defines the feature of childhood-onset SLE immunopathology (Figure 2) [31]. Similarly, another study explains that LDGs are characterized by spontaneous NETosis, upregulated expression of type I interferon (IFN)-stimulated genes, and metabolic dysregulation, including impaired redox capacity. This dysregulation exacerbates ROS-mediated tissue damage and accelerates disease severity [32,33]. For example, LDGs show increased interferon-stimulated protein level and altered phosphorylation of cytoskeletal regulators in SLE that dysregulate cellular trafficking and encourage retention within lung and vascular tissues. Biomechanically, LDGs exhibit rougher cell surfaces, and membrane perturbations are independent of immune complex exposure, suggesting that intrinsic cellular abnormalities and microvascular injury can be used as a retention marker in future [34].

Functionally, LDGs contribute to vasculopathy and sustained IFN production by stimulating plasmacytoid dendritic cells. Compounding this problem, impaired NET clearance due to the presence of anti-DNase I autoantibodies leads to an accumulation of extracellular DNA, worsening tissue injury [35] (Figure 2). A particularly concerning subset includes immature neutrophils (CD10^−^ CD16^−^), which are elevated in active SLE. These developmentally arrested cells exhibit poor phagocytosis but secrete inflammatory cytokines and undergo spontaneous NETosis. They can influence adaptive immune responses by modulating T cell activity [36]. Conversely, aged neutrophils express high CD62L, gather at the site of chronic inflammation, and cause T cell dysregulation, impaired migration, upregulated adhesion, and NETs, amplifying the inflammatory loops and vascular injury consecutively [37]. Moreover, excess B cell-activating factor (BAFF) release stimulated through proinflammatory neutrophils encourage abnormal B cell activation and autoantibody production. This process may induce B cell dysregulation through immune complex formation and IFNγ-mediated signaling [38]. These findings underscore the critical role of neutrophil subset diversity, including LDGs, immature, and aged neutrophils, in the immunopathogenesis of SLE. These subsets act as mediators of inflammation and represent valuable biomarkers and potential therapeutic subjects for monitoring disease.

### 3.2. Rheumatoid Arthritis (RA)

Neutrophils exhibit both significant heterogeneity as well as functional plasticity under the influence of cytokine factors that impact local anatomy in rheumatoid arthritis. Although proinflammatory subsets of neutrophil predominate the RA affected area, where ROS release was associated with pathology earlier, recent studies suggest their dual function. Here, lower ROS levels aggravate the inflammation through Th1/Th17 responses in the early stages of immune activation while elevated ROS levels in later stages are implicated in driving synovial tissue damage, ROS complex that present stage-specific pathogenesis of RA [39]. A subset of LDNs, elevated in RA, may be characterized by enhanced NETosis, an aberrant release of NETs composed of self-DNA, myeloperoxidase (MPO), and proteinase 3 (PR3) that directly contributes to synovial tissue damage (Figure 2). Inhibition of NETosis reduces the severity of disease through inducing potential anti-inflammatory pathways. However, NET and LDN inhibitors, Baricitinib, Tofacitinib, and Upadacitinib, have been reported to have a positive effect on inflammation in a Phase 3 clinical trial (Table 2) [40,41]. LDGs in RA also exhibit reduced apoptosis and diminished responsiveness to TNF-α, resulting in impaired chemotaxis, phagocytosis, ROS generation, and NET formation compared to NDNs. These functional deficits contribute to chronic inflammation and immune dysregulation [42]. Type I interferon (IFN-α)-regulated neutrophils further enhance RA pathogenesis. They show increased ROS production, delayed apoptosis via p38 MAPK, and chemokine reprogramming through the STAT signaling axis, functions distinct from neutrophils under normal physiological conditions [43]. A transcriptomic study has identified the miR-183C cluster as a dominant IFN-response gene signature across neutrophil subsets in RA. This microRNA cluster appears activated by circulating NETs and neutrophil-derived debris, potentially through the cGAS-STING pathway, which is known to amplify inflammatory cascades. This study strongly suggests that miR-183C could serve both as a therapeutic target (via inhibition) and a diagnostic/prognostic biomarker for rheumatoid arthritis [44]. In summary, the expanding knowledge of neutrophil heterogeneity and functional reprogramming in RA, while shaping disease outcomes, modulates neutrophil activity while preserving essential host defense mechanisms.

### 3.3. Juvenile Idiopathic Arthritis (JIA)

In juvenile idiopathic arthritis (JIA), certain neutrophils called LDNs, along with genes related to neutrophil activation, are found at higher levels in the blood. This increase leads to the release of effector molecules like myeloperoxidase (MPO), neutrophil elastase (NE), and MMP8, which can contribute to joint damage, similar to what is seen in RA. Studies also show that LDNs in JIA have impaired suppressive immune response, which may lead to ongoing inflammation. Interestingly, unlike in lupus (SLE), specific markers on the surface of neutrophils such as CD62L, CD66b, and CD11b are linked to lower levels of joint inflammation. This suggests that neutrophils in JIA may behave differently from those in other autoimmune diseases. However, the expression profile of these neutrophil subsets still requires validation as a differential marker of JIA, from other inflammatory conditions [55,56].

Further evidence highlights a sex-specific immune signature in systemic JIA (sJIA). Immature neutrophils and activation-associated genes are more prominent in female sJIA patients, indicating sexual dimorphism in neutrophil biology. Treatment with IL-1 receptor antagonists such as anakinra has proven effective in modulating these immature neutrophils [57]. In oligoarticular JIA, neutrophils in synovial fluid acquire an aged, activated phenotype. These cells show increased levels of CD206 and CD14 markers, usually found on monocytes, and lower levels of CD62L. This change in their surface markers is linked to reduced phagocytic ability and lower production of ROS, both of which are important for clearing cellular debris. As a result, these defects may contribute to ongoing joint inflammation and poor immune regulation in JIA. Increased CD206^+^ neutrophils with impaired phagocytosis and oxidative burst could represent a novel target for therapeutic intervention in persistent disease [58]. Additionally, functionally activated, proinflammatory neutrophils expressing elevated levels of IL-1β and TNF-α are implicated in driving synovial inflammation. These neutrophils may also influence adaptive immune responses, further contributing to disease severity and progression [59].

Evidence from autoimmune disease brings out plasticity of neutrophils contributing pathogenesis, which is equally evident in cancer. In autoimmunity, subsets like LDGs, immature, and aged neutrophils amplify inflammation via excessive NETosis, ROS generation, and autoantigen exposure. Similarly, tumor-associated neutrophils as well as stromal cues reprogram the neutrophils into proinflammatory and immunosuppressive phenotypes in oncology that enhance immunity and promote angiogenesis or metastasis, respectively. Despite opposite outcomes, for immune hyperactivation in autoimmune diseases versus suppression in cancer, both share common mechanisms, including NET signaling, type I interferon responses, metabolic remodeling, and immune cell crosstalk, under neutrophil-dependent immunity.

## 4. Neutrophils Heterogeneity in Cancer

### 4.1. Anti-Tumorigenic Neutrophils

N1 tumor-associated neutrophils (N1-TANs) are a specialized group of neutrophils that play an important role in fighting tumors within the TME. Unlike other neutrophil subsets, N1-TANs have strong tumor-killing (cytotoxic) abilities. They produce high levels of proinflammatory molecules such as TNF-α and ICAM-1, which help stimulate immune responses. Additionally, N1-TANs effectively induce T cell activation, further strengthening the host anti-tumor defense. They are typically induced in milieu where type I interferons are high and TGF-β is low, or under TGF-β blockade, to favor their anti-tumorigenic polarization [60]. Recent single-cell transcriptomic analyses have identified a unique subset of anti-tumor neutrophils in the TME marked by high HLA-DR and CD74 expression, enabling antigen presentation. This unique feature of these neutrophils can be influenced by leucine-driven epigenetic changes, especially an increase in a specific histone modification called H3K27ac. This modification turns on genes involved in antigen presentation and triggers the release of chemokines like CCL5, which helps attract T cells to the tumor site. In addition, these neutrophils produce inflammatory cytokines such as TNF-α and IFN-γ, which boost the immune response and support eliminating tumor cells through programmed cell death (apoptosis). Their presence correlates with improved patient survival, and they are distinguished from pro-tumor neutrophils across several metastatic cancers, including non-small-cell lung cancer (NSCLC), colorectal cancer, head and neck squamous cell carcinoma, melanoma, and triple-negative breast cancer (TNBC) [61]. Moreover, programmed cell death, a mitochondria-targeted tumor killing, has been highlighted recently which showed that anti-tumor neutrophils internalize into cancer cells via neuropilin-1 (NRP1)-mediated endocytosis by proteolytic cleavage of CD95/Fas receptor at specific sites (V220/A221 and I331/Q332) through effective neutrophil elastase (ELANE) and liberating a C-terminal death domain fragment (DDELANE). Further, this fragment binds to highly expressed histone H1 isoforms (e.g., H1.0, H1.2), and forms a DDELANE–H1 complex that translocate to the mitochondria. Consequently, it triggers mitochondrial dysfunction, elevated ROS production, DNA damage (γH2AX), and activation of the intrinsic apoptotic cascade (e.g., caspase-3, cleaved PARP), which enables H1-targeted selective killing of cancer cells. Additionally, ELANE-mediated tumor cell death releases tumor antigens, enhancing dendritic cell activation and promoting CD8^+^ T cell priming, which target distant metastases through a systemic anti-tumor immune response (Figure 3) [62]. Although N1-TANs share only a small portion of the total neutrophil population in tumors like NSCLC and colorectal cancer, their role as antigen-presenting cells and in activating T cells highlights their importance. These functions remark on the subset’s dynamic and inducible nature as their potential targets for neutrophil-based immunotherapies across various tumor types.

### 4.2. Pro-Tumorigenic Neutrophils

Pro-tumor neutrophils are commonly identified as N2-tumor-associated neutrophils (N2-TANs) or polymorphonuclear myeloid-derived suppressor cells (PMN-MDSCs) and the activated PMN-MDSC subset in cancer. Unlike anti-tumor neutrophils, pro-tumor neutrophils express low HLA-DR and CD74; rather, they exhibit Programmed death-ligand 1 (PDL-1), CD66b, and CD15, which contribute to immunosuppression in the tumor microenvironment [63]. A study revealed the elevated expression of immune suppressor genes (such as S100a8, S100a9, Mmp8, and Cybb) and inflammation (including Il1b, Socs3, and Ptgs2) as a result of pro-tumor neutrophils. Notably, CD14high neutrophils is an undisguisable tumor marker and exhibits the strongest suppressive activity. It suppresses T, B, and NK cell functions and promotes tumor progression through both immune and nonimmune mechanisms. The gene signature of tumor-infiltrating PMN-MDSCs shared similarity among mice and human PMN-MDSCs but found underscores to clinical outcomes in cancer patients [64]. The TME is profoundly affected by TGF-β, hypoxia, and tumor-derived cytokines, including IL-8 and G-CSF, which promote neutrophil polarization towards a pro-tumorigenic phenotype. Recently, a single-cell transcriptomic study across multiple cancer types highlighted vascular endothelial growth factor (VEGFA) and SPP1 (osteopontin) secretion as distinct transcriptional signatures. Pre-metastatic niches are nurtured through NET, VEGF, and MMP-9 secreted from N2-TANs that promote angiogenesis, epithelial-to-mesenchymal transition (EMT), and tumor cell intravasation. It suppresses T cell proliferation and cytotoxicity by upregulating arginase-1, ROS, and expressing immune checkpoint molecules like PD-L1, which implies immunosuppression as a hallmark (Figure 3) [61]. For example, breast ductal carcinoma in situ (DCIS) tissues have been shown to harbor more neutrophils than normal breast tissues. As these lesions transition into invasive ductal carcinomas, the TME becomes enriched in PD-L1-expressing neutrophils, regulatory T cells (Tregs), and dysfunctional TCR clonotypes, reflecting a deeply immune-suppressed landscape [65]. Similarly, in late-stage head and neck cancers, infiltrating neutrophils in TME are likely to be aged and possess suppressive characteristics, in contrast to those found at earlier tumor stages and mediate anti-tumor responses, initially [66].

### 4.3. Phenotypes Transition in TME: Anti-Tumorigenic (N1) and Pro-Tumorigenic (N2)

The tumor microenvironment is inclusion of local cues, abundant with cytokines, immune cells and other molecules that may regulate neutrophil polarization among anti-tumor (N1) and pro-tumor (N2) neutrophils. Adaptation into another subset provides plasticity in their function that determines the cancer outcomes as either progression or suppression by affecting related processes such as metastasis, angiogenesis, and immune regulation. For example, TANs can be polarized into either an anti-tumor ‘N1’ phenotype by enhancing immune activation and tumor control or a pro-tumor ‘N2’ phenotype by promoting immune suppression and tumor growth.

Transforming growth factor-beta (TGF-β) is reported as a switch to neutrophil phenotypic shift. TGF-β supports the pro-tumor N2 polarization by promoting angiogenesis through VEGF secretion, enhancing metastatic potential, and suppressing immune activation by upregulating PD-L1 expression, ultimately establishing an immunosuppressive tumor niche [61].

Blocking TGF-β signaling or introducing acute inflammatory stimuli can reprogram TANs from the N2 to the anti-tumor N1 phenotype (Figure 3). This shift may be marked by upregulation of immune activation genes, increased ROS-mediated cytotoxicity, enhanced recruitment, and activation of CD8^+^ T cells and natural killer (NK) cells. Restoration of antigen presentation led to direct tumor cell killing [15].

## 5. Therapeutic Targeting of Neutrophils

Advances in understanding neutrophil function, diversity, and their interactions with other immune cells have opened new possibilities for treating autoimmune diseases and cancers. Therapies are now being developed to reduce tissue damage by targeting NETosis in autoimmune conditions and to block neutrophil-driven inflammation in tumors using chemokine receptor inhibitors like CXCR2 antagonists. Many of these strategies are in preclinical or early clinical trials, showing promise in controlling autoimmune flares and improving cancer immunotherapy. Also, biomarkers such as low-density neutrophils (LDNs) and NET components are being studied for their potential in diagnosis, disease monitoring, and treatment feedback. Modulating the pathological subsets of neutrophil, without impairing host defense, may allow discovering neutrophil-targeted immunomodulation in both autoimmune and oncologic contexts.

The therapeutic landscape for neutrophil-targeted interventions reveals a contrary context across autoimmune diseases and cancer. In autoimmune diseases, strategies focus on dampening hyperactivated neutrophils through PAD4 inhibitors to prevent NETosis, DNase therapies to clear existing NETs, and JAK/STAT or IL-6 receptor blockade to reduce proinflammatory subsets like low-density neutrophils (Table 2). On the other hand, cancer therapeutics employ CXCR1/2 antagonists to block neutrophil recruitment and prevent their conversion to immunosuppressive myeloid-derived suppressor cells, to restore anti-tumor immunity (Table 2). Notably, agents such as JAK inhibitors show efficacy across both contexts, suggesting shared pathological mechanisms and opportunities for drug repurposing [67]. The future challenge lies in developing specific approaches that selectively target pathological neutrophil functions while preserving essential host defense roles. This leaves an important question on how biomarker-driven therapeutic strategies can be developed that may allow prediction and selective modulation of neutrophil phenotypes to switch them from pathogenic to homeostatic neutrophil, utilizing their plasticity, without compromising their fundamental antimicrobial and tissue repair functions.

## 6. Conclusions

Recent discoveries on neutrophil diversity, plasticity, and crosstalk with other immune cells contribute significantly in reshaping their multifaceted role in cancer and autoimmune diseases. Once recognized as an early responder, short-lived cells, neutrophils, are now known for their dynamic functions, either immuno-protective or pathogenic. Selective modulation in autoimmune diseases potentially minimizes chronic inflammation through targeting neutrophil-derived mediators like ROS, NETs, or BAFF. In cancer, reprogramming pro-tumor neutrophils to anti-tumor TANs presents a promising avenue for immunotherapy enhancement. Focus on identifying and validating neutrophil-based biomarkers (e.g., LDGs, NET components) for early diagnosis, disease monitoring, and predicting treatment response would be the future of this field.

## Figures and Tables

**Figure 1 ijms-26-09040-f001:**
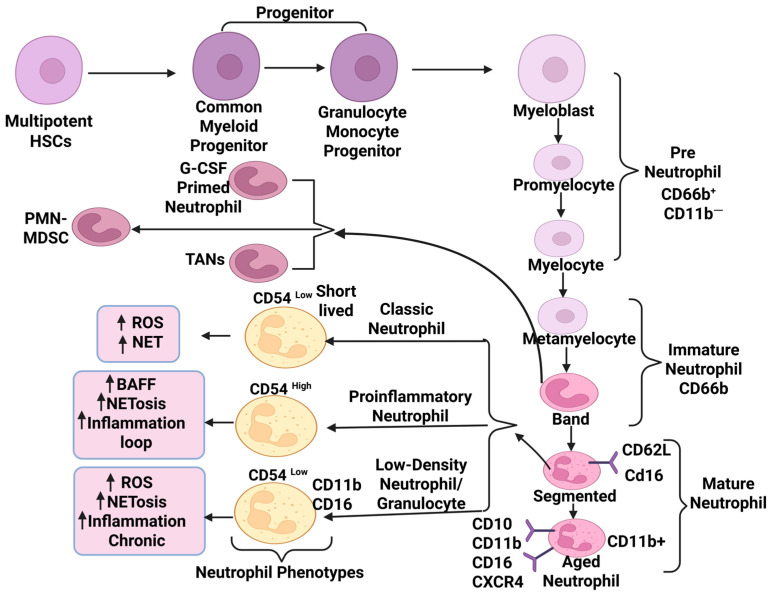
Diagrammatic representation of neutrophil development, differentiation into major subsets, and their later functions involved in autoimmune diseases and cancer. The figure highlights phenotypically and functionally distinct surface marker-defined population of neutrophils, including CD54^high^ mature neutrophils showing elevated (represented as an upright arrow) NETosis and reverse trans-endothelial migration. CD11b^high^ aged subsets exhibit increased adhesion, migration, and dysregulated activation under chronic inflammatory conditions. Similarly, low-density neutrophils (LDNs) or low-density granulocytes (LDGs) present immunosuppressive activity (T cell suppression), proinflammatory loop, NETosis, and marker profiles such as CD11b and CD16; CD66b banded immature neutrophil subsets arise into tumor-associated neutrophils (TANs), PMN-MDSC, and G-CSF-primed neutrophils under diseased conditions that display elevated adhesion such as cancer.

**Figure 2 ijms-26-09040-f002:**
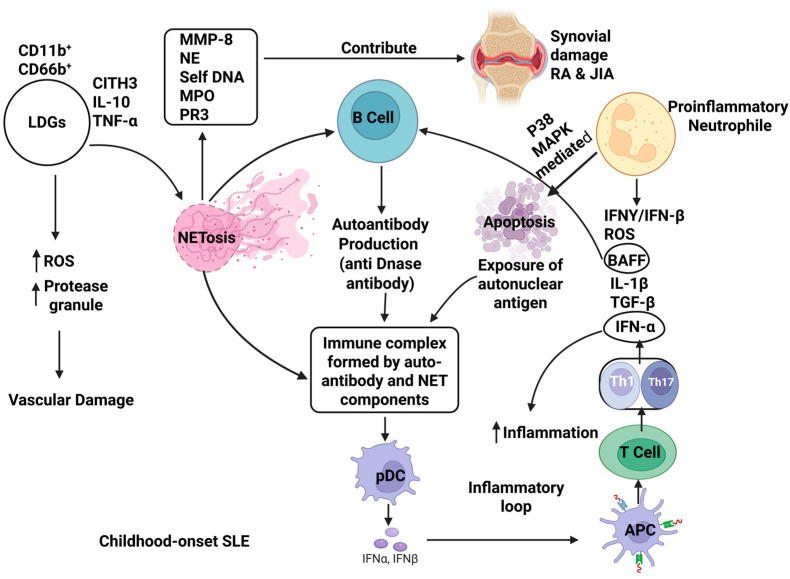
Schematic representation for role of low-density neutrophils and proinflammatory neutrophils in development of chronic inflammation childhood-onset SLE, RA, and JIA. LDGs releasing CitH3, IL-10, TNF-α, mediating high ROS (represented as an upright arrow) induced vascular damage and NETosis. Release of cytoplasmic protein, self-DNA, and cytokines during NETosis; promoting synovial damage in RA and JIA; BAFF-mediated B cell dysregulation; immune complex formation and type I IFN-mediated immune cell (pDC, APC, T cell) crosstalk and inflammatory loop formation through proinflammatory neutrophils. Abbreviations: SLE—systemic lupus erythematosus; RA—rheumatoid arthritis; JIA—juvenile idiopathic arthritis; LDGs—low-density granulocytes; CitH3—citrullinated histone H3; IL-10—Interleukin 10; TNF-α—tumor necrosis factor-α; NETosis—Neutrophil Extracellular Trap formation; type I IFN—interferon (such as IFN-α, IFN-β); ROS—reactive oxygen species; BAFF—B cell-activating factor of the tumor necrosis factor family; pDC—plasmacytoid dendritic cells; APC—antigen-presenting cells.

**Figure 3 ijms-26-09040-f003:**
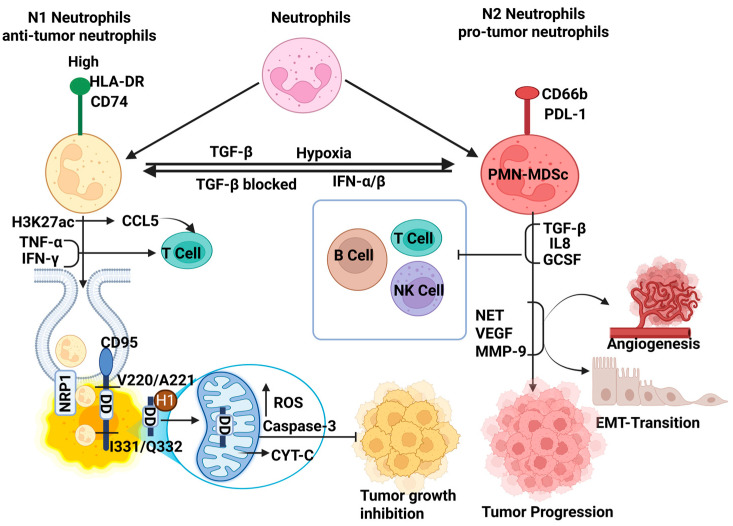
Schematic representation of anti-tumorigenic neutrophils and pro-tumorigenic and their transition vice versa. Anti-tumor neutrophils releases cytokines (CCL5, TNFα, IFNγ) that activate T cells and assists NRP1-mediated mitochondria-targeted tumor killing (anti-tumor neutrophils enter cancer cells via NRP1-mediated endocytosis, where effective neutrophil elastase cleaves CD95/Fas at specific sites (V220/A221 and I331/Q332) to release a C-terminal death domain fragment). This binds histone H1 isoforms, forming a complex that translocate to mitochondria, inducing dysfunction, ROS production (represented as an upright arrow), DNA damage, and activation of the intrinsic apoptotic cascade, enabling selective cancer cell killing. Pro-tumor neutrophil release TGF-β, IL-8, GCSF that inhibit immune cells (T, B, NK cells) activation, and promote tumor growth through angiogenesis as well as EMT. Transition among anti-tumor neutrophil to pro-tumor neutrophil under milieu rich in TGF-β and vice versa in TGF-β blockade condition. Abbreviations: NRP1—neuropilin-1; TGF-β—transforming growth factor-beta; IL-8—Interleukin-8; GCSF—Granulocyte colony-stimulating factor; PMN-MDSC—polymorphonuclear myeloid-derived suppressor cells.

**Table 1 ijms-26-09040-t001:** Description on transition among phenotypes of tumor-associated neutrophils.

S. No.	Transition of TANs	Key Factors	Phenotypic Markers	References
1.	Pro-tumor (N2) to Anti-tumor (N1)	Microbial signals, TGF-β blockade, IFN-α/β, acute inflammation	Increased ROS, HLA-DR^high^, CD74^high^, CD80, CD86, high chemokines recruiting T/NK cells	[15,16]
2.	Anti-tumor (N1) to Pro-tumor (N2)	TGF-β, chronic inflammation, hypoxia	Increased VEGF, PD-L1, ARG1, HLA-DR^low^ and CD74^low^	[16,17]

**Table 2 ijms-26-09040-t002:** Description of neutrophil-targeted, drugs, or therapeutic agents in autoimmune diseases and cancer.

Neutrophil-Targeted Therapeutics in Autoimmune Diseases and Cancer (Clinical Trials)				
Drugs and Therapeutic Agents	Mechanism of Action	Phase/Status of Clinical Trial	Outcome of Therapeutics	References
Baricitinib, Tofacitinib, Upadacitinib	Modulate neutrophil activation and NETosis targeting JAK/STAT pathway	Phase 1–3 trials	Reduces NETs, LDNs	[45] ClinicalTrials.gov ID NCT02535689 ClinicalTrials.gov ID NCT05843643
Taxifolin	Inhibits NETosis (Nrf2)	Preclinical/ex vivo	Protective in lupus models	[46]
Dual-acting DNase1/DNase1L3	Degrades NET DNA	Preclinical	Promising in SLE models	[47]
Metformin	Inhibits NETosis	Add-on trial (completed)	Reduced flares, indirect NETosis effect	[48]
Anifrolumab	Type I Interferon Inhibitor	Approved, real-world/clinical	Reduces NETs and LDNs; rapid efficacy in SLE	[49]
JBI-589 Peptidylargininedeiminase 4 (PAD4) inhibitor	Inhibits PAD4; blocks NETosis	Preclinical (animal models, in vitro human/mouse neutrophils)	Highly selective PAD4 inhibitor; blocks NET formation and reduces arthritis severity in mouse models; confirmed NET inhibition in vito	[50]
Tocilizumab	IL-6R blockade; modulates neutrophil function/NETs	Phase 3, approved	Reduces NETosis, neutrophil infiltration, and improves neutrophil function in RA, systemic JIA	[51] REC reference 10/H0904/14
AZD5069	Inhibits CXCR2	Phase 1/2 (combo with immunotherapy)	Inhibits neutrophil recruitment to tumors; reduces intratumoral neutrophil and MDSC infiltration, shown to decrease neutrophil levels and improve response in Prostate, NSCLC, solid tumors	[52] ClinicalTrials.gov ID NCT02583477
Navarixin (SCH-527123, MK-7123)	Inhibits CXCR2	Phase 1b/2	Inhibits CXCR2-mediated neutrophil chemotaxis and tumor infiltration; reduces tumor-promoting neutrophils in NSCLC, solid tumors	[53,54] ClinicalTrials.gov ID NCT03473925
Reparixin	Inhibits CXCR1/2	Phase 2	Blocks neutrophil recruitment and MDSC infiltration by antagonizing CXCR1/2 in breast and pancreatic cancer	[53,54]

## Data Availability

No new experimental data were created.

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
