# Peer review of "Roles of Neutrophils in Autoimmune Diseases and Cancers"

_ijms, 2025, doi:10.3390/ijms26189040_

Round 1
Reviewer 1 Report
Comments and Suggestions for Authors
Firstly, I would like to express my gratitude to the authors for creating such a remarkable and thorough review on the diverse roles of neutrophils in autoimmune diseases and cancer. This work represents a significant commitment to gathering and synthesizing a vast amount of literature, merging mechanistic insights with translational viewpoints, and presenting the information in a well-organized and accessible way. The complete coverage, ranging from neutrophil biology and phenotypic diversity to disease-specific mechanisms and potential therapeutic strategies, will ensure that this review becomes an invaluable reference for both researchers and clinicians.
The manuscript provides a comprehensive and well-structured review of neutrophil biology, functional plasticity, and heterogeneity in the context of autoimmune diseases and cancer. It effectively synthesizes recent advances regarding neutrophil subsets, their microenvironment-driven polarization, and various therapeutic targeting strategies. The treatment of disease-specific contexts, such as systemic lupus erythematosus, rheumatoid arthritis, and juvenile idiopathic arthritis, as well as tumor-associated neutrophil phenotypes (N1/N2), is commendable. Moreover, the integration of mechanistic and translational perspectives enhances its relevance for a diverse audience in the fields of immunology and oncology.
Several strengths merit particular recognition:
- The thorough integration of current literature across both autoimmunity and cancer domains.
- A clear and accessible explanation of neutrophil subtypes, including their surface markers and functional characteristics.
- Well-illustrated diagrams that effectively convey mechanistic pathways.
- The inclusion of therapeutic strategies pertinent to preclinical and clinical environments.
That said, a few refinements could enhance the manuscript further:
1. Clarification of Terminology for Low-Density Neutrophils and Granulocytes: The terms low-density neutrophils(LDNs) and low-density granulocytes (LDGs) appear to be used interchangeably at times. It would be beneficial to clearly define these terms early in the manuscript to prevent any potential confusion. For instance, LDNs are characterized by their reduced buoyant density, which causes them to separate with PBMCs during density-gradient centrifugation. This population is heterogeneous and may include immature neutrophils, activated mature neutrophils, and pathologically altered subsets. LDGs represent a functionally distinct pro-inflammatory subset of LDNs, most commonly characterized in autoimmune diseases by enhanced NETosis, elevated type I interferon–stimulated gene expression, and increased pro-inflammatory cytokine release. This clarification is vital for interpreting the disease-specific sections and understanding the relationship between these populations.
2. Enhancement of Figure Legends for Interpretation: Figures 1–3 are indeed informative; however, they contain several abbreviations that are not defined in the captions. Expanding the figure legends to include definitions for all abbreviations and brief explanations of key processes would greatly facilitate understanding independently of the main text.
3. Bridging the Autoimmune and Cancer Sections: The transition between the sections on autoimmune diseases and cancer feels somewhat abrupt. A brief linking paragraph could emphasize that in both contexts, neutrophil phenotypes are shaped by microenvironmental cues yet lead to divergent biological outcomes, such as immune amplification in autoimmunity versus immune suppression in cancer.
4. Synthesis of Therapeutic Trends from Table 2, which offers excellent detail on drug mechanisms, trial statuses, and outcomes. Including a short narrative synthesis after the table would help highlight key therapeutic patterns. For example, discussing these therapeutic perspectives in the section would enhance the impact of the great work.
- For autoimmune diseases, therapies often concentrate on inhibiting NETosis (PAD4 inhibitors, DNase therapies) or modulating neutrophil activation (JAK inhibitors, IL-6R blockade).
- In cancer treatment, strategies more frequently target chemokine receptors (CXCR1/2) to mitigate tumor infiltration by pro-tumor neutrophils and myeloid-derived suppressor cells (MDSCs).
- Notably, some therapies, such as JAK inhibitors, may have relevance across both contexts.
5. Current Unresolved Research Questions: While the review is thorough, including a brief forward-looking statement on open questions, such as the stability of N1/N2 phenotypes in vivo, the overlap between LDGs and PMN-MDSCs, and the potential for therapeutic approaches that span different diseases, would add significant value as a reference for future research endeavors. For example, How stable or reversible are N1/N2 phenotypes in vivo? What is the exact overlap and distinction between LDGs and PMN-MDSCs in different diseases? Can neutrophil-targeted therapies be designed to work across both autoimmune and cancer contexts without impairing host defense? Adding these points would position the review as a forward-looking resource for future research.
Overall Recommendation: This review is excellently written, very timely, and thorough. Addressing the points above would further improve the manuscript's clarity, cohesion, and analytical depth, ensuring it serves as both a detailed reference and a valuable guide for ongoing research in neutrophil biology, autoimmunity, and cancer. I thank the authors once again for their significant contribution to the field.
Author Response
Reply to Reviewer 1:
Comment 1. Clarification of Terminology for Low-Density Neutrophils and Granulocytes: The term slow-density neutrophils (LDNs) and low-density granulocytes (LDGs) appear to be use d interchangeably at times. It would be beneficial to clearly define these terms early in the manuscript to prevent any potential confusion. For instance, LDNs are characterized by their reduced buoyant density, which causes them to separate with PBMCs during density gradient centrifugation. This population is heterogeneous and may include immature neutrophils, activated mature neutrophils, and pathologically altered subsets. LDGs represent a functionally distinct pro-inflammatory subset of LDNs, most commonly characterized in autoimmune diseases by enhanced NETosis, elevated type I interferon–stimulated gene expression, and increased pro-inflammatory cytokine release. This clarification is vital for interpreting the disease-specific sections and understanding the relationship between these populations.
Response- We acknowledge your suggestion very much, which are of great importance
in improving our manuscript. We have added, “Low-density granulocytes (LDGs), also referred to as low-density neutrophils (LDNs), are a proinflammatory neutrophil subset that co-exists with peripheral blood mononuclear cells during density gradient centrifugation and is enriched in various immune-mediated diseases [22]” in line 123 to 125.
Herein, Low-density neutrophils (LDNs) is an operational term that refers to neutrophils (or neutrophil-lineage cells) that co-fractionate with peripheral blood mononuclear cells (PBMCs) during density-gradient centrifugation due to reduced buoyant density. The LDN fraction is heterogeneous and can contain immature neutrophils, activate mature neutrophils (for example degranulated or density-altered cells), and pathologically alter neutrophil subsets. Low-density granulocytes (LDGs) are a functionally defined subset commonly identified within the LDN fraction; LDGs are enriched for a pro-inflammatory phenotype that has been most extensively described in autoimmune diseases. LDGs typically display enhanced NETosis, upregulated type I interferon-stimulated gene signatures, and increased pro-inflammatory cytokine production. Thus, while “LDN” describes a density-based isolation outcome, “LDG” denotes a pro-inflammatory granulocytic subset frequently observed within that fraction. We have used these terms throughout the manuscript with this distinction in mind.
Comment 2: Enhancement of Figure Legends for Interpretation: Figures 1–3 are indeed informative; however, they contain several abbreviations that are not defined in the captions. Expanding the figure legends to include definitions for all abbreviations and brief explanations of key processes would greatly facilitate understanding independently of the main text.
Response- Thank you for your insightful comment. Key information for figure 1 and abbreviations for figure 2 (as it was already defined) and details as well as abbreviation for figure 3 has been added to the part of legend in illustrated figures in the revised manuscript.
Figure 1: “The figure highlights phenotypically and functionally distinct surface marker-defined population of neutrophils including CD54high mature neutrophils shows elevated NETosis and reverse trans-endothelial migration. CD11bhigh aged subsets exhibit increased adhesion, migration, and dysregulated activation under chronic inflammatory conditions. Similarly, low-density neutrophils (LDNs) or low-density granulocytes (LDGs) presents immunosuppressive activity (T cell suppression), proinflammatory loop, NETosis, and marker profiles such as CD11b and CD16; CD66b banded immature neutrophil subsets arises into tumor-associated neutrophils (TANs), PMN-MDSC and G-CSF–primed neutrophils under diseased condition that displays elevated adhesion such as cancer.”
Figure 2: “Abbreviations: SLE- Systemic Lupus Erythematosus, RA- Rheumatoid Arthritis, JIA- Juvenile Idi-opathic Arthritis, LDGs- Low Density Granulocytes, CitH3- Citrullinated histone H3, IL-10- In-terleukin 10, TNF-α-Tumor Necrosis Factor-α, NETosis- Neutrophil Extracellular Trap formation, Type-I IFN- Interferon (such as IFN-α, IFN-β), ROS- Reactive oxygen species, BAFF-B-cell-activating factor of the tumor-necrosis-factor family, pDC- plasmacytoid dendritic cells, APC-Antigen presenting cells.”
Figure 3: “(Antitumor neutrophils enter cancer cells via NRP1-mediated endocytosis, where effective neu-trophil elastase cleaves CD95/Fas at specific sites (V220/A221 and I331/Q332) to release a C-terminal death domain fragment. This binds histone H1 isoforms, forming a complex that translocate to mitochondria, inducing dysfunction, ROS production, DNA damage, and activation of the intrinsic apoptotic cascade, enabling selective cancer cell killing); Abbreviation: NRP1- Neuropilin-1, TGF-β- Transforming Growth Factor beta, IL-8- Interleukin-8, GCSF- Granulocyte Colony-Stimulating Factor, PMN-MDSC- Polymorphonuclear myeloid-derived suppressor cells.”
Comment 3: Bridging the Autoimmune and Cancer Sections: The transition between the sections on autoimmune diseases and cancer feels somewhat abrupt. A brief linking paragraph could emphasize that in both contexts, neutrophil phenotypes are shaped by microenvironmental cues yet lead to divergent biological outcomes, such as immune amplification in autoimmunity versus immune suppression in cancer.
Response- We acknowledge your suggestion very much, which are of great importance
in improving our literature. We have added a connecting paragraph following your suggestion in line 277 to 286. “Evidences from autoimmune disease bring out plasticity of neutrophils contributing pathogenesis, which is equally evident in cancer. In autoimmunity, subsets like LDGs, immature, and aged neutrophils amplify inflammation via excessive NETosis, ROS generation, and autoantigen exposure. Similarly, tumor associated neutrophils as well as stromal cues reprogram the neutrophils into proinflammatory and immunosup-pressive phenotypes in oncology that enhance immunity and promote angiogenesis or metastasis, respectively. Despite opposite outcomes, immune hyperactivation in au-toimmune diseases versus suppression in cancer, both shares common mechanisms, including NET signaling, type I interferon responses, metabolic remodeling, and im-mune cell cross-talk, under neutrophil dependent immunity.”
Comment 4: Synthesis of Therapeutic Trends from Table 2, which offers excellent detail on drug mechanisms, trial statuses, and outcomes. Including a short narrative synthesis after the table would help highlight key therapeutic patterns. For example, discussing these therapeutic perspectives in the section would enhance the impact of the great work.
- For autoimmune diseases, therapies often concentrate on inhibiting NETosis (PAD4inhibitors, DNase therapies) or modulating neutrophil activation (JAK inhibitors, IL-6Rblockade).
- In cancer treatment, strategies more frequently target chemokine receptors (CXCR1/2) to mitigate tumor infiltration by pro-tumor neutrophils and myeloid-derived suppressor cells (MDSCs).
- Notably, some therapies, such as JAK inhibitors, may have relevance across both contexts.
Response- We acknowledge your suggestion very much, which are of great importance
in improving our manuscript. We have added, in line 402 to 412 within manuscript. “The therapeutic landscape for neutrophil-targeted interventions reveals a contrary context across autoimmune diseases and cancer. In autoimmune diseases, strategies focus on dampening hyperactivated neutrophils through PAD4 inhibitors to prevent NETosis, DNase therapies to clear existing NETs, and JAK/STAT or IL-6 receptor blockade to reduce proinflammatory subsets like low-density neutrophils (Table 2). In other hand, cancer therapeutics employ CXCR1/2 antagonists to block neutrophil recruitment and prevent their conversion to immunosuppressive myeloid-derived sup-pressor cells, to restore anti-tumor immunity (Table 2). Notably, agents such as JAK inhibitors show efficacy across both contexts, suggesting shared pathological mechanisms and opportunities for drug repurposing [67]. The future challenge lies in developing specific approaches that selectively target pathological neutrophil functions while preserving essential host defense roles.”
Comment 5: Current Unresolved Research Questions: While the review is thorough, including a brief forward-looking statement on open questions, such as the stability of N1/N2 phenotypes in vivo, the overlap between LDGs and PMN-MDSCs, and the potential for therapeutic approaches that span different diseases, would add significant value as a reference for future research endeavors. For example, how stable or reversible are N1/N2 phenotypes in vivo? What is the exact overlap and distinction between LDGs and PMN-MDSCs indifferent diseases? Can neutrophil-targeted therapies be designed to work across both autoimmune and cancer contexts without impairing host defense? Adding these points would position the review as a forward-looking resource for future research.
Response- Thank you for your insightful explanations and cues to add open question we acknowledge your suggestion very much, which are of great importance in improving our manuscript. We have added a precise key question on development of biomarkers and its utility in both autoimmune diseases and cancer, in between line 412 – 417. “This remain an important question that how biomarker-driven therapeutic strategies can be develop that may allow prediction and selective modulation of neutrophil phenotypes to switching them from pathogenic to homeostatic neutrophil utilizing their plasticity, without compromising their fundamental antimicrobial and tissue repair functions?”
Reviewer 2 Report
Comments and Suggestions for Authors
This is a very well-written and comprehensive review that provides valuable insights into neutrophil biology. A few suggestions for further strengthening the manuscript:
- A recent perspective published in Immunity (PMID: 40763729) may be relevant for expanding the discussion on neutrophil heterogeneity and classification. The authors are encouraged to incorporate insights from this work to enhance the depth of the review.
- While CD11b is commonly used as a marker for immature monocytes, it is important to clarify that it is not exclusively expressed on neutrophils. The authors may wish to address this point to avoid any potential misinterpretation and acknowledge the overlapping expression across myeloid subsets.
- The inclusion of key transcription factors involved in neutrophil development would significantly enrich the mechanistic understanding provided in the review. Discussing factors such as C/EBP family members, PU.1, and Gfi1 could provide important context.
- In my opinion, it would be valuable to briefly highlight some of the advanced tools and methodologies currently used to dissect neutrophil function and plasticity, such as single-cell RNA sequencing, mass cytometry, and high-dimensional imaging approaches.
Overall, this review makes a strong contribution to the field and will be of broad interest to readers studying innate immunity and myeloid cell biology.
Author Response
Reply to Reviewer 2:
Comment 1: A recent perspective published in Immunity (PMID: 40763729) may be relevant for expanding the discussion on neutrophil heterogeneity and classification. The authors are encouraged to incorporate insights from this work to enhance the depth of there view.
Response- We thank you to recommend this review from cell press highlighting the recent perspective in neutrophil Immunity. This work emphasizes the need for a clear and standardized system to classify neutrophil heterogeneity, considering their maturation stage, location in tissues, and functional adaptations which is awesome. Although our review focuses on autoimmune diseases and cancer, where we have explained underlie subsets and their plasticity using recent research articles. we agree that such a framework would help bring clarity to the many neutrophil subsets described in these diseases. Including standardized criteria could make it easier to compare results across studies and support the development of therapies that specifically target disease-driving neutrophil populations instead we have reported therapeutics went through clinical trials and their mechanism with an open question on enhancement in therapeutics against autoimmune disease and cancer in revised manuscript.
Comment 2: While CD11b is commonly used as a marker for immature monocytes, it is important to clarify that it is not exclusively expressed on neutrophils. The authors may wish to address this point to avoid any potential misinterpretation and acknowledge the overlapping expression across myeloid subsets.
Response- We acknowledge your suggestion regarding CD11b expression. While CD11b is frequently used as a marker in studies of neutrophil activation and adhesion, it is not specific to neutrophils and is also expressed on other myeloid cells, including monocytes, macrophages. In the revised manuscript, we have clarified this point to avoid misinterpretation and have noted the importance of using CD11b in combination with additional markers when identifying neutrophil populations. We have modified the sentence between line 102 to 105 that is “Another surface marker, CD11b, represents active adhesion and migration, expressed by myeloid cells including neutrophil, macrophage and monocytes. It resides stably within secretory granules of neutrophils and is expressed at a minimal level on the surface along with other neutrophil markers such as CD16”.
Comment 3: The inclusion of key transcription factors involved in neutrophil development would significantly enrich the mechanistic understanding provided in the review. Discussing factors such as C/EBP family members, PU.1, and Gfi1 could provide important context.
Response- We thank you for this valuable suggestion. We agree that transcription factors such as members of the C/EBP family, PU.1, and Gfi1 play critical roles in neutrophil development. While our review is primarily focused on the role of neutrophil heterogeneity and plasticity in autoimmune diseases and cancer rather than development of neutrophil because these transcription factors involve in expressing distinct granulocytes from myeloid progenitor cells such as neutrophil which remain unchanged among different subsets of neutrophil profound in pathology condition as well as homeostatic condition. Here, we have focused in different subsets of neutrophil that co-exists in autoimmune diseases and cancer.
Comment 4: In my opinion, it would be valuable to briefly highlight some of the advanced tools and methodologies currently used to dissect neutrophil function and plasticity, such as single-cell RNA sequencing, mass cytometry, and high-dimensional imaging approaches.
Response- We thank you for this insightful suggestion. We agree that advanced technologies such as single-cell RNA sequencing, mass cytometry, and high-dimensional imaging have greatly advanced our understanding of neutrophil function and plasticity. However, as our review specifically focuses on the roles of neutrophils in autoimmune diseases and cancer, an in-depth methodological discussion falls outside the intended scope. Nevertheless, we have mentioned techniques such single cell transcriptomics and relevance of their analysis that bring valuable output to dissect neutrophil function and plasticity with reference of research articles. As it enabling more precise characterization of neutrophil subsets, which will likely enhance disease-focused research in the future.